

# Multiple immunity-related genes control susceptibility of *Arabidopsis thaliana* to the parasitic weed *Phelipanche aegyptiaca*

Christopher R. Clarke[1], So-Yon Park[2], Robert Tuosto[2], Xiaoyan Jia[2], Amanda Yoder[3], Jennifer Van Mullekom[3] and James Westwood[2]

[1] Genetic Improvement of Fruits and Vegetables Laboratory, United States Department of Agriculture, Agricultural Research Service, Beltsville, MD, USA
[2] School of Plant and Environmental Sciences, Virginia Tech, Blacksburg, VA, USA
[3] Department of Statistics, Virginia Tech, Blacksburg, VA, USA

## ABSTRACT

Parasitic weeds represent a major threat to agricultural production across the world. Little is known about which host genetic pathways determine compatibility for any host–parasitic plant interaction. We developed a quantitative assay to characterize the growth of the parasitic weed *Phelipanche aegyptiaca* on 46 mutant lines of the host plant *Arabidopsis thaliana* to identify host genes that are essential for susceptibility to the parasite. *A. thaliana* host plants with mutations in genes involved in jasmonic acid biosynthesis/signaling or the negative regulation of plant immunity were less susceptible to *P. aegyptiaca* parasitization. In contrast, *A. thaliana* plants with a mutant allele of the putative immunity hub gene *Pfd6* were more susceptible to parasitization. Additionally, quantitative PCR revealed that *P. aegyptiaca* parasitization leads to transcriptional reprograming of several hormone signaling pathways. While most tested *A. thaliana* lines were fully susceptible to *P. aegyptiaca* parasitization, this work revealed several host genes essential for full susceptibility or resistance to parasitism. Altering these pathways may be a viable approach for limiting host plant susceptibility to parasitism.

## INTRODUCTION

The parasitic weed *Phelipanche aegyptiaca* (syn. *Orobanche aegyptiaca*) is an obligate holoparasite, lacking the capacity for photosynthesis and fully dependent on parasitization of a host plant for nutrients and completion of its lifecycle. *P. aegyptiaca*, which is commonly known as Egyptian broomrape, is a major biotic constraint to crop production throughout much Eastern Europe, Asia and Northern Africa (*Parker, 2013*). *P. aegyptiaca* is able to parasitize a remarkably broad host range of dicotyledonous plants including crop and non-crop species. The crop-containing plant families for which *P. aegyptiaca* is a major threat are *Solanaceae, Fabaceae, Apiaceae* and *Cucurbitaceae* (*Parker, 2013*). Additionally, *P. aegyptiaca* can successfully parasitize the model host plant *Arabidopsis*

Corresponding authors
Christopher R. Clarke,
christopher.clarke@usda.gov
James Westwood, westwood@vt.edu

*thaliana* which makes it a promising model parasitic plant (*Goldwasser, Plakhine & Yoder, 2000*; *Westwood, 2000*), despite the significant challenges of working with *P. aegyptiaca* in a laboratory setting.

*Phelipanche aegyptiaca* is a member of the Orobanchaceae family, which includes the overwhelming majority of agriculturally relevant parasitic weeds. Other members of the family that are major pests in agricultural production systems include *Striga hermonthica* (purple witchweed), a widespread and devastating parasite of cereal crops throughout Africa (*Spallek, Mutuku & Shirasu, 2013*), *Orobanche cumana* (sunflower broomrape), one of the primary biotic constraints to sunflower production in Europe and Asia (*Molinero-Ruiz et al., 2015*) and *Orobanche cernua* (nodding broomrape), a costly weed of tomato and other solanaceous crops throughout Africa, Asia and Europe (*Parker, 2013*). Although much has been written about the ability of parasitic weeds to germinate in response to host-specific chemical cues (*Zwanenburg, Pospíšil & Ćavar Zeljković, 2016*), develop haustoria to invade host plant tissues (*Yoshida et al., 2016*), and extract nutrient through vascular connections (*Irving & Cameron, 2009*), relatively little is known about the host plant immune responses and parasitic plant virulence mechanisms throughout these stages of plant–parasite interactions (See *Kaiser et al. (2015)* and *Clarke et al. (2019)* for review).

Because of the close physical association and shared angiosperm characteristics between parasitic plants and their hosts, mechanical and chemical controls are largely ineffectual in the control of parasitic weeds. Development of genetically resistant host crops is the most promising strategy for managing parasitic weeds (*Rubiales, Rojas-Molina & Sillero, 2016*). Unfortunately, to date only a few resistance (*R*) genes or resistance-associated quantitative trait loci have been identified for parasitic weeds. A classic nucleotide-binding-site, Leucine-rich-repeat (LRR) *R* gene that confers resistance to *S. gesnerioides* was cloned from cowpea (*Li & Timko, 2009*). A pattern recognition receptor (PRR) in tomato that confers resistance to the stem parasite *Cuscuta reflexa* was identified in tomato (*Hegenauer et al., 2016*). Recently, another LRR receptor-like kinase was identified as responsible for sunflower resistance to *O. cumana* (*Duriez et al., 2019*). Several other resistance loci have been deployed in the management of *O. cumana* on sunflower, but the parasite rapidly overcomes such resistance (*Molinero-Ruiz et al., 2015*).

An alternative genetic strategy for the control of parasitic weeds is the alteration of host plant genes that are essential for parasite attachment or development, so called susceptibility genes (*Van Schie & Takken, 2014*). For example, genes involved in the biosynthesis of parasite germination stimulants or transport of critical nutrients to the parasite are potential susceptibility genes. Identifying the genetic pathways that underpin host compatibility for parasite attachment and development is an essential first step toward finding susceptibility genes. To that end, we developed an assay to quantify the ability of *P. aegyptiaca* to successfully attach and develop on *A. thaliana* roots, and quantified susceptibility to parasite attachment and development for 46 mutant lines of *A. thaliana*. The primary goal in this work was to test the impact on a parasitic plant of host plant hormone signaling and immunity-related genes that have been either
demonstrated or hypothesized to be involved in plant susceptibility to other pathogen classes.

## MATERIALS AND METHODS

### Plant growth and quantifying parasite attachment and development

The parasite attachment assay was conducted using a randomized incomplete block design due to large number of tested genotypes. For each experimental block, four mutant plant lines were randomly selected from the pool of all mutant lines in the collection. Approximately 25 *A. thaliana* seeds of each of the four mutant lines plus the wild type background ecotype were stratified in water at 4 °C for 2 days in then planted in Sunshine #1 potting mix (Sungro, Agawam, MA, USA). Plants were grown in a Conviron ATC40 growth chamber at 20 °C, 12-h light cycle and light intensity of 90 μmol m²s⁻¹ for 10 days. Eight Polyethylene (PE) bags with glass fiber grade A (GFA) paper (Whatman, Maidstone, UK) backings were made as previously described (*Westwood, 2000*) with dimensions of 26 cm × 9 cm (Fig. 1A). The 10-day-old *A. thaliana* seedlings were gently removed from the soil and the roots were washed with water until free of soil. The plants were transplanted such that the roots were positioned between the PE bag and the GFA paper and the hypocotyl extended from the top of the bag (Fig. 1A ). One plant of each of the four mutant lines and one plant of the wild type background ecotype were randomly distributed into each of the PE bags and placed under a laboratory growth light with a 12-h light cycle for 12 days. *P. aegyptiaca* seeds were sterilized following previously described protocols (*Westwood, 2000*) and placed on wet GFA paper in a sealed Petri plate (Fisher Scientific, Waltham, MA, USA) for six days to condition the seeds (*Westwood, 2000*). The strigolactone analog rac-GR24 was then applied to the conditioned *P. aegyptiaca* seeds at a concentration of 2 mg/L to stimulate germination. The Petri dishes were resealed and stored in the dark for an additional 24 h. Conditioned and stimulated *P. aegyptiaca* seeds were inoculated onto the roots of the *A. thaliana* plants in the PE bags (12 days after transplanting from soil) using a fine-tipped paintbrush. By placing germinated *P. aegyptiaca* seeds immediately adjacent to developed roots, the impact of differential root development among the various tested *Arabidopsis* mutants was limited. The seeds were aligned to be within 0.3 mm of roots that had grown since the transplanting (lighter colored roots) and approximately 35 seeds were placed along the roots of each plant in each bag.

The attachment rate for each individual *A. thaliana* plant was quantified 15 days after inoculation. Every *P. aegyptiaca* seed was examined under a dissecting microscope (Zeiss Stemi SV11) at 12×g magnification and classified as either non-germinated, germinated but not attached, or attached. Not attached and attached germinated seeds were distinguished by probing with a 0.3 mm wide dissecting probe to test for adherence of the parasite radicle to the *A. thaliana* roots. The attached *P. aegyptiaca* plants were further classified as either early-stage attachment/haustoria connection, early-stage tubercle, or late-stage tubercle which correspond to the stages 3, 4.1 and 4.2 in the Parasitic Plant Genome Project datasets (*Westwood et al., 2012*; *Yang et al., 2014*) (Fig. 1D). The 0.3 mm dissecting probe was used to distinguish the three attached classifications as follows: an

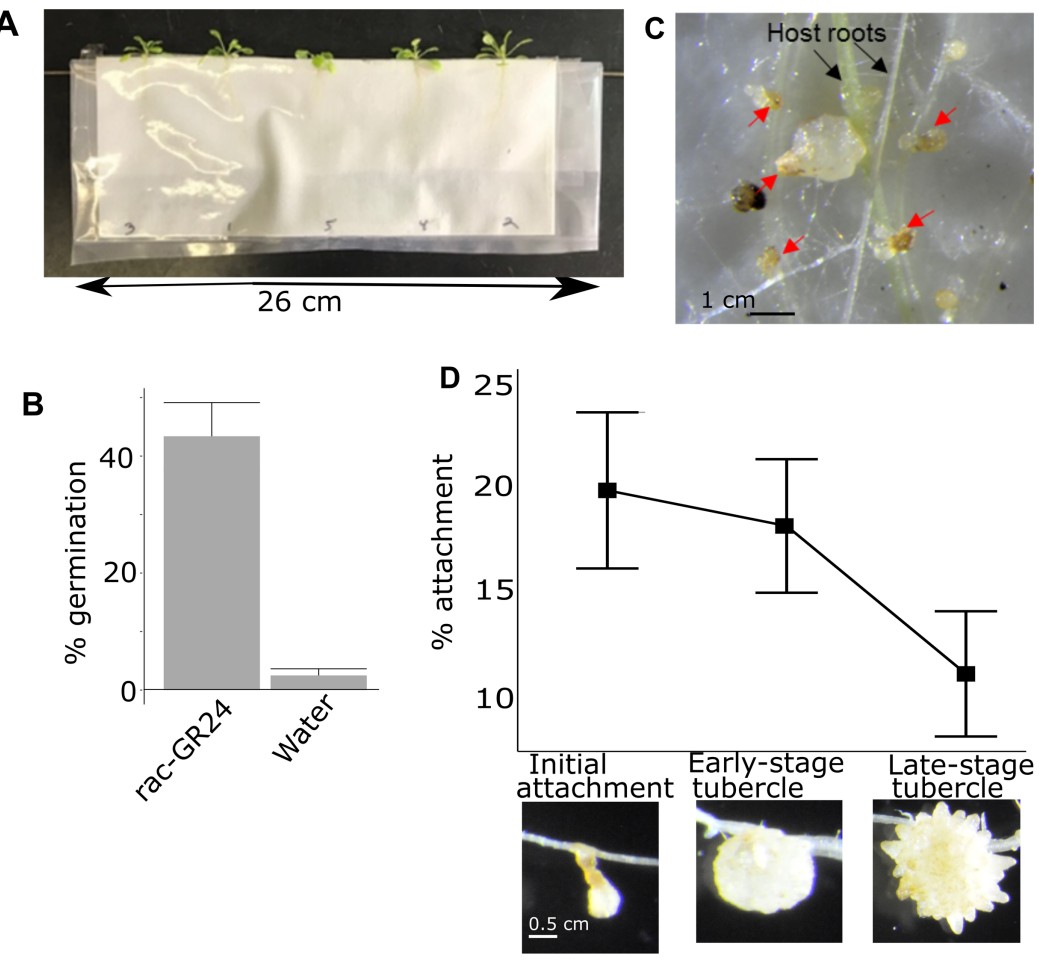

**Figure 1 Pipeline for quantifying attachment and tubercle development rate of *P. aegyptiaca* on *A. thaliana*.** (A) Example of *A. thaliana* lines growing in PE bags before inoculation with *P. aegyptiaca*. Plants were randomly distributed. (B) Typical germination rates of *P. aegyptiaca* seedlings following conditioning and treatment with germination stimulants *n* = 3 plates for each germination stimulant with approximately 100 total seeds per plate. (C) Example of germinated and ungerminated *P. aegyptiaca* radicles (red arrows) on *A. thaliana* roots. Germinated seedlings are counted to quantify the attachment and tubercle development rates. (D) Rate of attachment and tubercle development of *P. aegyptiaca* on wildtype (Col) *A. thaliana* roots for the three parasite development stages considered in this work—initial haustorial connection and attachment, early-stage tubercle, and late-stage tubercle *n* = 8. The attachment rate is the number of seeds that reached that stage of development divided by the total number of *germinated* seeds.               

attached radicle thinner than 0.3 mm was categorized as an attached radicle/initial attachment (stage 3). A tubercle formed that is thicker than 0.3 mm but has no secondary roots was categorized as an early-stage tubercle (stage 4.1) and coincides with the completion of the vascular connection/feeding bridge between the parasite and the host plant. A tubercle with secondary roots longer than 0.3 mm was categorized as a late-stage tubercle (stage 4.2). The rate of attachment for each of the three stages was determined by calculating the ratio of all of the *P. aegyptiaca* seeds that had reached at least the designated stage (i.e., the rate of initial attachment included the count of late-stage tubercles plus early-stage tubercles) divided by the total number of germinated

*P. aegyptiaca* seeds (i.e., the total number of all stages plus those seeds that germinated but did not attach to hosts).

## Statistical analysis of attachment rates

We used a generalized linear mixed model (*SAS/STAT(R) 9.2 User's Guide SE, 2016*) to analyze different rates of parasitization at each of the three developmental stages. This model is based on the binomial distribution with the logit link. The SAS script and example files are uploaded to the Ag Data Commons (https://data.nal.usda.gov/dataset/data-multiple-immune-pathways-control-susceptibility-arabidopsis-thaliana-parasitic-weed-phelipanche-aegyptiaca). The one fixed effect of interest was treatment of the plants. The treatments are the different host plant genetic pathways which were compared to the respective wildtype in our model (Col-0, Ler-0, or Ws-2). Both bag and position were set as fixed effects to account for any systematic variability within both bag and position. The experiment was specified as a random block, which allows us to determine significance of the fixed effects above and beyond experiment to experiment variability. This also accounts for the fact that results within an experiment may be more related than results across experiments. Our response variable was the ratio of attachment stage total to overall germinated seed total (described above). We compared 95% and 99% confidence intervals on the odds ratio estimates at each of the three stages to determine which treatments are statistically significantly different from the wildtype. An odds ratio greater than one indicates that *P. aegyptiaca* is more successful on the mutant line relative to the wildtype. An odds ratio less than one indicates that *P. aegyptiaca* is less successful on the mutant line than on the wildtype.

Type III tests of fixed effects indicated that there was no significant variation dependent on the bag. However, there was a significant effect dependent on the position of the plant within the bag for early and late stage tubercle attachment rates (Table S1). Tukey–Kramer analysis of the least square means of the differences of position with adjustment for multiple comparisons showed that significant differences ($p < 0.05$) always included plants on the edge of the bag (positions 1 and 5). We hypothesize that these differences were due to the edge of the GFA paper drying out more unevenly than the center of the GFA paper. Therefore, it is essential to randomize the position of the plants within each bag, as was done here.

## Time-lapse photography

A PE bag containing *A. thaliana* eco. Col-0 was inoculated with stimulated *P. aegyptiaca* seeds as described above. The bags were mounted on a metal hanger inside a 11.3-L Rubbermaid container that had a 70 mm diameter hole cut in the side for insertion of a macro camera lens and the top portion was covered with aluminum foil with a narrow opening so that the *A. thaliana* shoots could be exposed to light (12-h light-dark cycles) while the parasites were in darkness. The camera was a Sigma SD14 digital camera with a 70 mm Sigma DG macro lens. The camera flash provided the light source for the photography. The *A. thaliana* was watered through a tube that fed directly into the bag, so that the plant did not have to be disturbed during the time-lapse photography.

For recording aboveground plant growth, a tomato (*Solanum lycopersicum*) was used as host because *Arabidopsis* does not support robust aboveground growth in our experimental setup. A Nikon D5100 camera with a Nikon DX AF-S Nikkor lens was used. Pictures were taken at 9-min intervals for 37 days over the course of the above and below ground segments. Sigma Photo Pro software (Sigma Corporation of America, Ronkonkoma, NY, USA) was used to convert the images from raw X3F format into JPEG. Adobe Premiere Pro CC 2017 was then used to assemble the images and for video editing to make the time-lapse video at 24 fps.

## Quantitative RT-PCR

Three biological replicates of approximately 40 0.5 cm root sections were harvested from both inoculated and time-matched mock-inoculated *A. thaliana* eco. Col-0 plants at early attachment stage, early vascularization/early tubercle stage, and late tubercle stage (six total experimental conditions with three biological replicates and three technical replicates each). RNA was extracted using a Qiagen RNeasy kit. cDNA was synthesized using Superscript cDNA synthesis kit (Invitrogen, Carlsbad, CA, USA) following the manufacturer's recommended protocol. qRT-PCRs were performed by StepOnePlus Real-Time PCR System (Invitrogen, Carlsbad, CA, USA) with Power SYBR green master mix (Invitrogen, Carlsbad, CA, USA). The reaction condition was 2 min at 50 °C and 10 min at 95 °C followed by 40 cycles at 95 °C for 15 s, at 56 °C for 15 s and 72 °C for 15 s. Dissociation curves were evaluated to confirm the specificity. Relative quantification was calculated by $2^{-\Delta\Delta Ct}$ method using qBase+ software. Three housekeeping genes (*SAND, UBQ10* and *UFP*) were used for the normalization. geNorm analysis revealed that geNorm M-value of all three genes (*SAND, UBQ10* and *UFP*) were 0.762, 0.709 and 0.619, relatively (qBASE+, Biogazelle). The recommended cutoff value is one or less. Calibrated, normalized relative quantification values for each gene were compared between the mock-inoculated and *Phelipanche*-inoculated plants for each parasite developmental stage using a two-tailed, two-sample unequal variance *t*-test. The primers used are described in Table S2. All primers were previously designed in other studies: (*Jacobs et al., 2011*) (*VSP2, MYB51, EXPPT1, BOI, SID2, WRKY53*), (*Paponov et al., 2008*) (*IAA13, IAA2, ACS6*), (*Nguyen et al., 2016*) (*ARR10, UBQ10*), (*Šašek et al., 2014*) (*SAND*), (*Tran, Chen & Wang, 2017*) (*UFP*), (*Pegadaraju et al., 2007*) (*PAD4*), (*Zhang et al., 2014*) (*PR1*). All primers were validated to be specific to *A. thaliana* DNA and to not amplify any product from cDNA prepared from early tubercle stage *P. aegyptiaca* tissue.

# RESULTS

## A quantitative screen for host plant resistance and susceptibility to parasitic weeds

We developed a quantitative assay to measure the susceptibility of *A. thaliana*–*P. aegyptiaca* attachment and tubercle development based on previously described techniques (*Westwood, 2000*). *A. thaliana* plants grown in PE bags with GFA backing (Fig. 1A) allow for host roots to be directly inoculated with *P. aegyptiaca*. All *P. aegyptiaca* seeds were stimulated with the strigolactone analog rac-GR24 (*Yoneyama et al., 2010*),

which is an effective germination stimulant for *P. aegyptiaca* (Fig. 1B). *P. aegyptiaca* attachment and development can be monitored in this growth system. Our analysis of time-lapse video of *P. aegyptiaca* (Video S1—available at https://figshare.com/articles/ PhelipancheParasitization_mov/11894388) growth highlighted the acceleration in parasite growth following development of the floral meristem. Therefore, the challenge for our assay of parasite growth on mutants was to devise a method that could provide sensitive, reproducible measures of parasite success during a developmental stage characterized by relatively small changes in growth. The number of *P. aegyptiaca* seedlings at various growth stages can be counted under a dissecting microscope (Fig. 1C). Graphical representation of the typical attachment and tubercle development rates from inoculation of *P. aegyptiaca* onto wildtype *A. thaliana* is shown in Fig. 1D. All data from all parasite attachment experiments are in Table S3. Because of the substantial number of comparisons made in this analysis (three different stages for each of 46 mutant lines compared to wildtype), we primarily focus on differences that met the more stringent 99% confidence limit cutoff (see "Methods") to minimize false positives. Based on this cutoff, only between 2 and 9 mutant lines showed altered susceptibility to *P. aegyptiaca* parasitization at any specific stage compared to wildtype *A. thaliana* (Table S4).

## Host genetic pathways essential for susceptibility to *P. aegyptiaca*

*Phelipanche aegyptiaca* is a successful pathogen of *A. thaliana* (*Goldwasser, Plakhine & Yoder, 2000*); therefore, our assay of attachment and development rates is more suited to identify mutations that attenuate *A. thaliana* susceptibility to *P. aegyptiaca* parasitization than mutations that increase susceptibility. The 46 tested mutant lines include numerous disruptions in key signaling pathways (Table 1). It is important to note that many of these genes are involved in several different genetic pathways due to the substantial crosstalk among many of the signaling pathways. Examples include *pad4* and *eds1* which are involved in salicylic acid (SA) signaling, but also critical for several *R* gene-mediated responses (*Cui et al., 2017*; *Rustérucci et al., 2001*) and the substantial crosstalk between jasmonic acid (JA), SA and ethylene signaling (*Koornneef & Pieterse, 2008*).

For the stage of initial parasitization/haustoria development, no mutant lines were less susceptible than wildtype *A. thaliana* at the 99% confidence level (Fig. 2). This result demonstrates that none of the tested host endogenous signaling pathways are essential for *P. aegyptiaca* to form haustoria and attach to host roots. However, plants overexpressing *Ethylene Response Factor 2* (*ERF2*), a transcription factor upregulated in the presence of ethylene that activates numerous immune responses (*Catinot, Huang & Zimmerli, 2015*), were significantly less susceptible than wildtype to *P. aegyptiaca* initial attachment at the 95% confidence level, suggesting that elevated host ethylene levels may attenuate initial attachment. The attachment rates were more variable within and among experiments for the early attachment stage compared to the two later stages measuring tubercle growth (Table S3).

Eight *A. thaliana* mutant lines were attenuated in the ability to support *P. aegyptiaca* early-stage tubercle formation at the 99% confidence limit (Fig. 3). Five of the less

**Table 1** The 46 mutant lines considered in this study.

| Gene | Involved in[1] | Background | Citation | ABRC # |
|---|---|---|---|---|
| 35s:ERF1-2 | Ethylene response/ JA signaling | Col | Lorenzo et al. (2003) | CS6143 |
| 35s:PMR4 | Penetration resistance | Col | Ellinger et al. (2013) | n/a |
| aba1 | ABA signaling | Col | Alonso et al. (2003) | SALK_027326C |
| arr1-2/arr10-1/arr11-1 | Cytokinin signaling | Ws-2 | Mason et al. (2005) | CS6993 |
| aux1-7 | Auxin distribution | Col | Pickett, Wilson & Estelle (1990) | CS3074 |
| aux1-7; ein2 | Auxin distribution/ethylene perception | Col | | CS8843 |
| axr1-3 | Auxin, cytokinin, JA, ethylene signaling | Col | Lincoln, Britton & Estelle (1990) | CS3075 |
| bak1-4 | Master regulator of pattern-triggered immunity | Col | Alonso et al. (2003); Kemmerling et al. (2007) | SALK_116202C |
| bkk1 | Master regulator of pattern-triggered immunity | Col | Alonso et al. (2003); He et al. (2007) | SALK_057955C |
| cpr5 | Negative regulator of systemic acquired resistance and programed cell death | Col/No-0 | Boch et al. (1998) | CS3770 |
| csn5 | Core targeted immunity hub | Col | Alonso et al. (2003); Mukhtar et al. (2011) | SALK_027705 |
| dcl1-7 | RNAi | Ler | Robinson-Beers, Pruitt & Gasser (1992) | CS3089 |
| dde2-2 | JA biosynthesis | Col | Von Malek et al. (2002) | CS65993 |
| dde2-2/ein2-1/pad4-1/sid2-2 | JA biosynthesis, ethylene signaling, SA signaling | Col | Tsuda et al. (2009) | CS66007 |
| dde2-2/pad4-1 | JA biosynthesis, SA signaling | Col | Tsuda et al. (2009) | CS65998 |
| dde2-2/sid2-2 | JA biosynthesis, SA signaling | Col | Tsuda et al. (2009) | CS65999 |
| edr1-1 | Ethylene-dependent stress responses | Col | Frye & Innes (1998) | CS67959 |
| eds1-1 | Negative regulator R gene-mediated resistance | Ws-2 | Parker et al. (1996) | n/a |
| ein2-1 | Ethylene signaling | Col | Alonso et al. (1999) | n/a |
| gai1 | Giberellic acid signaling | Ler | Koorneef et al. (1985) | CS63 |
| gai-t6 rga-t2 rgl1-1 rgl2-1 | DELLA genes, giberellic acid signaling | Col | Navarro et al. (2008) | n/a |
| jar1-1 | JA biosynthesis | Col | Staswick, Su & Howell (1992) | n/a |
| jar1-1/axr1-3 | JA signaling, auxin signaling | Col | Tiryaki & Staswick (2002) | CS67934 |
| jar1-1/mlo2-11 | JA signaling, penetration resistance | Col | Consonni et al. (2006) | CS9723 |
| jaz3/jaz4-1/jaz9-1 | JA signaling (repressor) | Col | | n/a |
| Jaz3 (T3) | JA signaling (overexpression Jaz3) | Col | | n/a |
| jin1-1 | Jasmonic acid signaling | Col | Berger, Bell & Mullet (1996) | n/a |
| lsd1-2 | Negative regulation of cell death and disease resistance | Col | Kaminaka et al. (2006) | CS68738 |
| lsu2-1 | Core targeted immunity hub | Col | Alonso et al. (2003); Mukhtar et al. (2011) | SALK_031648c |
| lsu2-2 | Core targeted immunity hub | Col | Alonso et al. (2003); Mukhtar et al. (2011) | SALK_126244c |
| mkk1/mkk2 | Map kinase signaling | Col | Qiu et al. (2008) | n/a |
| mlo2-5; pen2-1 | Penetration resistance | Col | Consonni et al. (2006) | CS9717 |
| mpk4 | Map kinase signaling | Col | Qiu et al. (2008) | n/a |
| NahG | SA antagonist | Col | Lawton (1995) | n/a |
| ndr1-1 | R gene-mediated resistance, systemic acquired resistance | Col | Century, Holub & Staskawicz (1995) | CS6358 |
| Gene | Involved in[1] | Background | Citation | ABRC # |
|---|---|---|---|---|
| pad4-1 | SA and SA-independent defense responses | Col | Glazebrook, Rogers & Ausubel (1996) | CS3806 |
| pad4-1/ndr1-1 | SA signaling, systemic acquired resistance, R gene-mediated resistance | Col | | n/a |
| pen2-3 | Penetration resistance | Col | Lipka et al. (2005) | CS66946 |
| pepr1/pepr2 | Perception of DAMPs | Col | Krol et al. (2010) | n/a |
| pfd6-1 | Core targeted immunity hub | Col | Alonso et al. (2003); Mukhtar et al. (2011) | n/a |
| rar1 | Master regulator of R gene-mediated immunity | Col | Muskett et al. (2002) | n/a |
| rdr1-1/rdr2-1/rdr6-15 | RNAi (known problems with pleiotropy) | Col | Garcia-Ruiz et al. (2010) | CS66485 |
| rdr6-11 | RNAi | Col | Peragine et al. (2004) | CS24285 |
| rpm1 | R gene-mediate immunity | Col | Grant et al. (1995) | n/a |
| sgt1 | Master regulator of R gene-mediate immunity | Col | Austin et al. (2002) | n/a |
| sid2-2 | SA biosynthesis | Col | Nawrath & Métraux (1999) | n/a |
| tir1 | Auxin perception | Col | Ruegger et al. (1998) | CS3798 |

**Note:**
[1] An incomplete representation of the pathways in which the gene is involved.

susceptible mutants involve knockouts in jasmonic acid biosynthesis or signaling (*dde2, dde2/pad4, dde2/sid2, jar1/mlo, jar1/axr1*) suggesting that *P. aegyptiaca* requires functional host JA signaling and biosynthesis to form successful tubercles. However, neither the *jar1* nor *jin1* single mutant is less susceptible to *P. aegyptiaca* tubercle formation. *pad4* plants were also less susceptible to *P. aegyptiaca* tubercle development. PAD4 is involved in both SA signaling and TIR *R*-gene signaling. Because other mutants defective in SA accumulation (*sid2* and *NahG*) and *R*-gene-mediated responses (*sgt1, rar1* and *rpm1*) were not significantly less susceptible to parasite attachment, the reason *pad4* plants are less susceptible at the early tubercle stage remains uncertain. Lastly, the mutants *cpr5* and *edr1* were substantially less susceptible to early-stage tubercle formation. Both of these mutants have been previously shown to constitutively express multiple plant defense pathways and be more resistant to several microbial biotrophic plant pathogens (*Bowling et al., 1997*; *Christiansen et al., 2011*; *Tang, Christiansen & Innes, 2005*; *Yoshida et al., 2002*).

Six of the mutants (*jar1/axr1, jar1/mlo, cpr5, dde2/pad4, dde2* and *dde2/sid2*) identified as being less susceptible to early-stage tubercles also supported significantly fewer late-stage *P. aegyptiaca* tubercles at the 99% confidence limit (Fig. 4). Additionally, the *jaz4/5/9* mutant, which is defective in JA responses because the JAZ4, JAZ5 and JAZ9 proteins are all transcriptional regulators triggered by jasmonate (*Pauwels & Goossens, 2011*; *Yan et al., 2014*), also supported fewer late-stage tubercles than WT *A. thaliana*. The JA biosynthesis mutant *jar1* supported fewer late stage tubercles at the 95% confidence level giving further support to the hypothesis that several genes involved in host JA signaling and biosynthesis are essential for *P. aegyptiaca* parasitism.

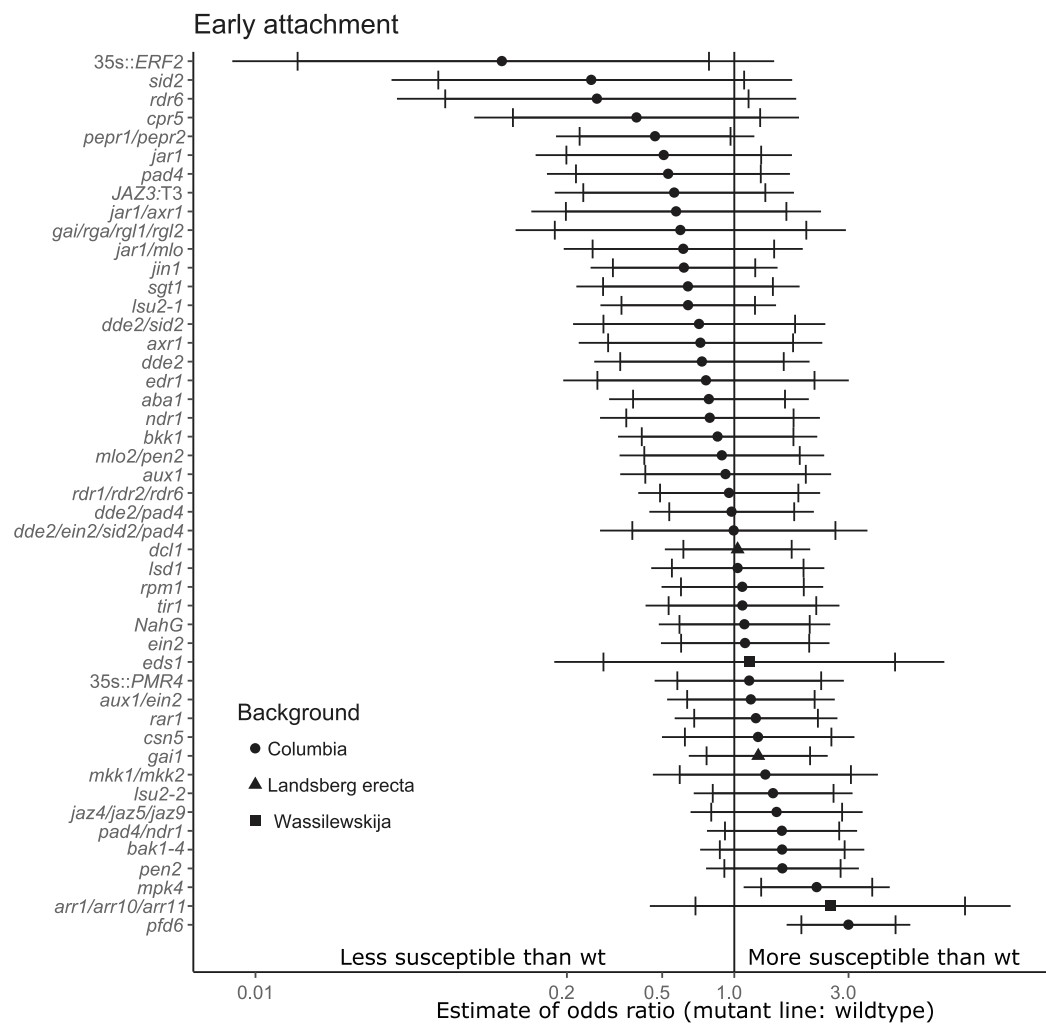

**Figure 2 Mutations in multiple immunity-related genes significantly affect the susceptibility of *A. thaliana* to initial parasitization by *P. aegyptiaca*.** Marker points indicate the estimated odds ratio of the rate of early attachment of *P. aegyptiaca* on the mutant line relative to wildtype *A. thaliana*. An odds ratio greater than one indicates that *P. aegyptiaca* is more successful on the mutant line than wildtype. An odds ratio less than one indicates that *P. aegyptiaca* is less successful on the mutant line than wildtype. The shapes of the marker points depict the ecotype background for each mutant line. The capped error bars represent the 95% confidence interval and the uncapped error bars (overlapping but extending) represent the 99% confidence interval. A confidence interval that does *not* cross the vertical line at attachment rate = 1 (the normalized attachment rate to wildtype *Arabidopsis*) is considered statistically different at the indicated confidence level. Data represent at least 14 replicates from at least two independent experiments for each mutant line.                 

In addition to the JA signaling and biosynthesis mutant lines, the DELLA quad mutant (*gai/rga/rgl1/rgl2*) and *rdr6* mutant lines supported fewer late-stage tubercles at the 99% confidence level. The DELLA genes (*GAI, RGR* and *RGL*s) are negative growth regulators controlled by Gibberellic acid (GA), and this quad mutant is more resistant to the biotrophic pathogen *Pseudomonas syringae* and more susceptible to the necrotrophic pathogen *Alternaria brassicicola* (*Navarro et al., 2008*).

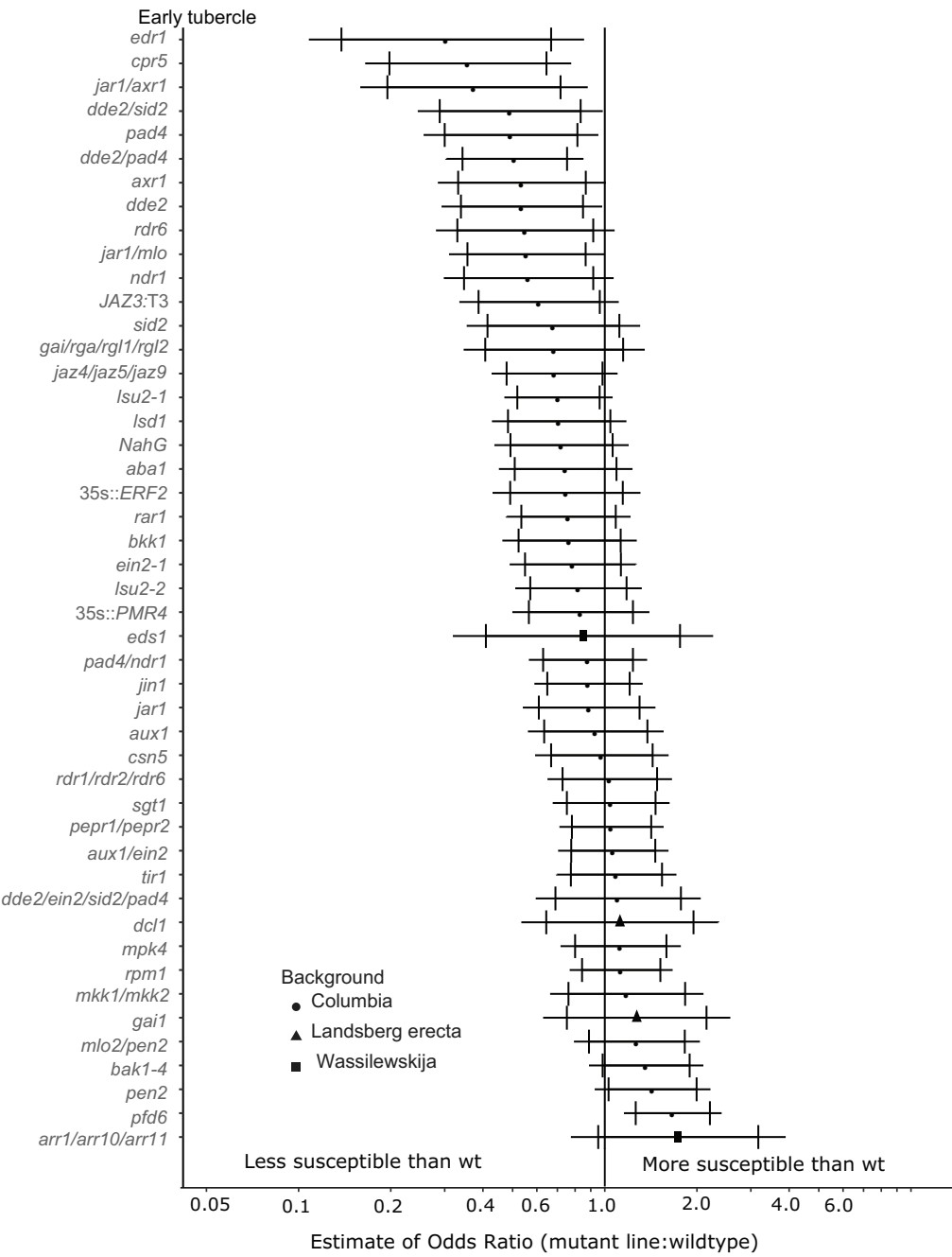

**Figure 3 Mutations in multiple immunity-related genes significantly affect the susceptibility of _A. thaliana_ to early tubercle development by _P. aegyptiaca_.** Marker points indicate the estimated odds ratio of the rate of early attachment of _P. aegyptiaca_ on the mutant line relative to wildtype _A. thaliana_. An odds ratio greater than one indicates that _P. aegyptiaca_ is more successful on the mutant line than wildtype. An odds ratio less than one indicates that _P. aegyptiaca_ is less successful on the mutant line than wildtype. The shapes of the marker points depict the ecotype background for each mutant line. The capped error bars represent the 95% confidence interval and the uncapped error bars (overlapping but extending) represent the 99% confidence interval. A confidence interval that does _not_ cross the vertical line at attachment rate = 1 (the normalized attachment rate to wildtype _Arabidopsis_) is considered statistically different at the indicated confidence level. Data represent at least 14 replicates from at least two independent experiments for each mutant line.     

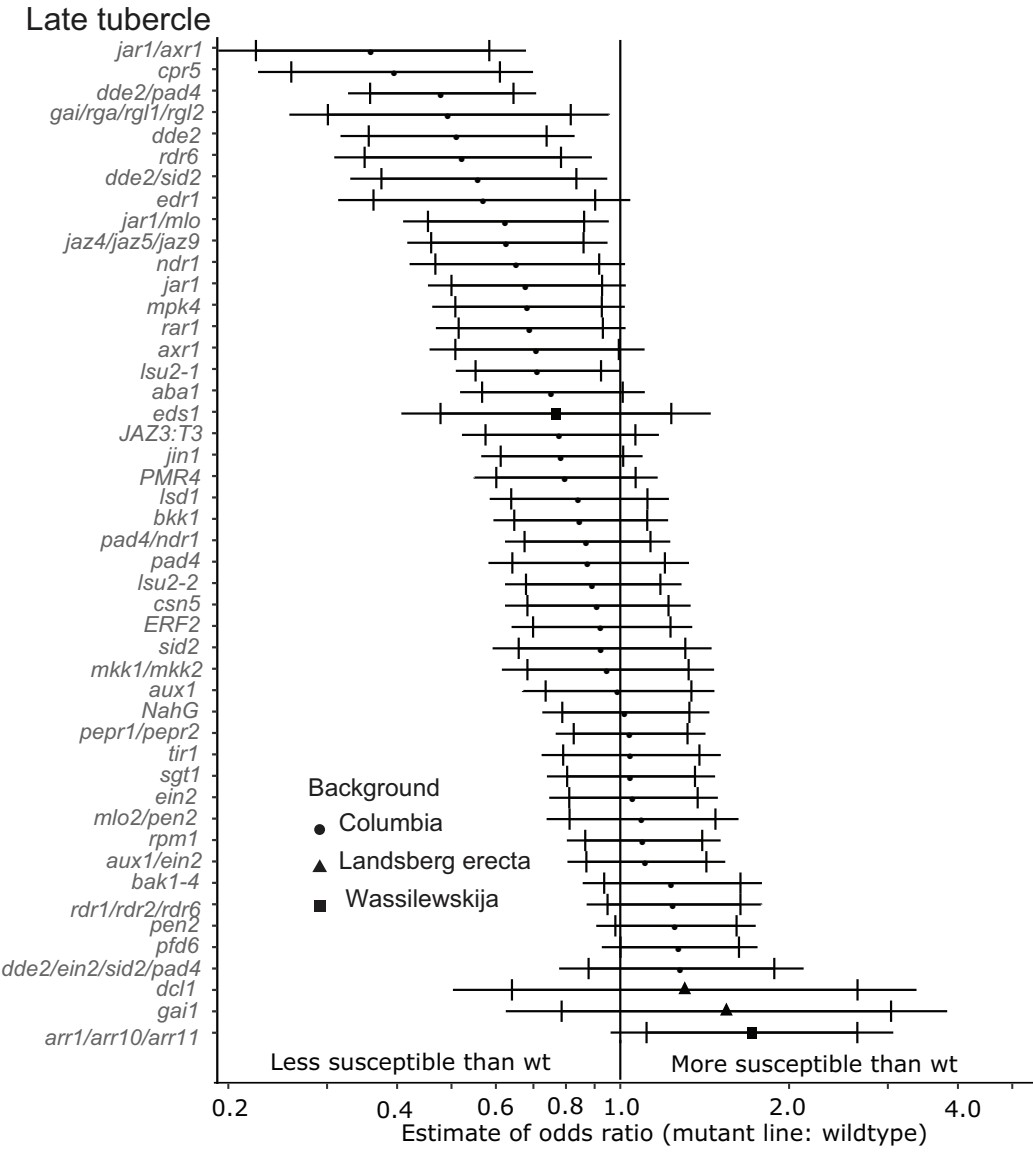

**Figure 4 Mutations in multiple immunity-related genes significantly affect the susceptibility of *A. thaliana* to supporting late tubercle development by *P. aegyptiaca*.** Marker points indicate the estimated odds ratio of the rate of early attachment of *P. aegyptiaca* on the mutant line relative to wildtype *A. thaliana*. An odds ratio greater than one indicates that *P. aegyptiaca* is more successful on the mutant line than wildtype. An odds ratio less than one indicates that *P. aegyptiaca* is less successful on the mutant line than wildtype. The shapes of the marker points depict the ecotype background for each mutant line. The capped error bars represent the 95% confidence interval and the uncapped error bars (overlapping but extending) represent the 99% confidence interval. A confidence interval that does *not* cross the vertical line at attachment rate = 1 (the normalized attachment rate to wildtype *Arabidopsis*) is considered statistically different at the indicated confidence level. Data represent at least 14 replicates from at least two independent experiments for each mutant line.

## Host genetic pathways essential for resistance to *P. aegyptiaca*

Two tested mutants, *mpk4* and *pfd6-1*, were more susceptible to initial *P. aegyptiaca* parasitization (Fig. 2). PFD6 is a subunit of the prefoldin complex, and *pfd6* mutants are

defective in microtubule function and formation (*Gu et al., 2008*) and is a putative hub of the *A. thaliana* immune system (*Mukhtar et al., 2011*). *pfd6-1* mutants were also significantly more susceptible to supporting early-stage tubercle development by *P. aegyptiaca* (Fig. 3). We, therefore, hypothesize that PFD6 is an important component of the host plant immune response against plant parasitization.

None of the tested mutant lines supported significantly more late-stage *P. aegyptiaca* tubercles compared to wildtype *A. thaliana* at the 99% confidence level (Fig. 4). However, *pfd6* did support significantly more late-stage tubers at the 95% confidence level. Additionally, the triple mutant *arr1/arr10/arr11*, which is deficient in cytokinin signaling (*Mason et al., 2005*), also supported more late-stage *P. aegyptiaca* tubercles at the 95% confidence level.

## Parasitization by *P. aegyptiaca* alters the transcription of several immunity-related genes

We monitored transcript levels of 12 defense-and hormone signaling-associated genes of *A. thaliana* (Table S2) following *P. aegyptiaca* parasitization. Significant upregulation of the glucosinolate biosynthetic gene *MYB51* and downregulation of the negative cell death regulator BOI were observed during initial haustorial attachment (Fig. 5). During early stage tubercle development, the pattern-trigged immunity (PTI) marker gene *WRKY 53* is significantly upregulated. During late-stage tubercle development, only smaller magnitude changes in gene expression were observed, most notably significant down-regulation in the JA marker gene *VSP2* and the SA marker gene *SID2*. Whether these changes are more directly associated with host defense responses or parasite virulence strategies remains unknown.

## DISCUSSION

The importance of several genes for *A. thaliana* resistance and susceptibility to *P. aegyptiaca* was revealed by our novel quantitative screen for plant parasitism. Given the artificial nature of the PE bag growth system, these observations are a starting point for elucidating the host molecular pathways that dictate whether a host is resistant or susceptible to attack from a parasitic plant. Notably, the *mpk4, gai1* and *cpr5* plants exhibited marked reduced growth compared to wildtype in the PE bags. Additionally, substantial crosstalk among many of the studied pathways (*Koornneef & Pieterse, 2008*), complicates our ability to definitively list which host pathways are critical for supporting plant parasitism. Nevertheless, this data provides the foundation for understanding the host genetics of susceptibility to plant parasitism. We discuss the host pathways that appear to be the most important below.

### JA/SA signaling

Several host genes involved in JA biosynthesis and signaling are (e.g., *dde2, jar1, jaz4/5/9*) are required for full host susceptibility to development of parasite tubercles (Figs. 3 and 4). Several of these mutants also supported fewer early-stage haustorial attachments (Fig. 2), but not significantly as determined by our stringent cutoff (Table S4). Therefore,

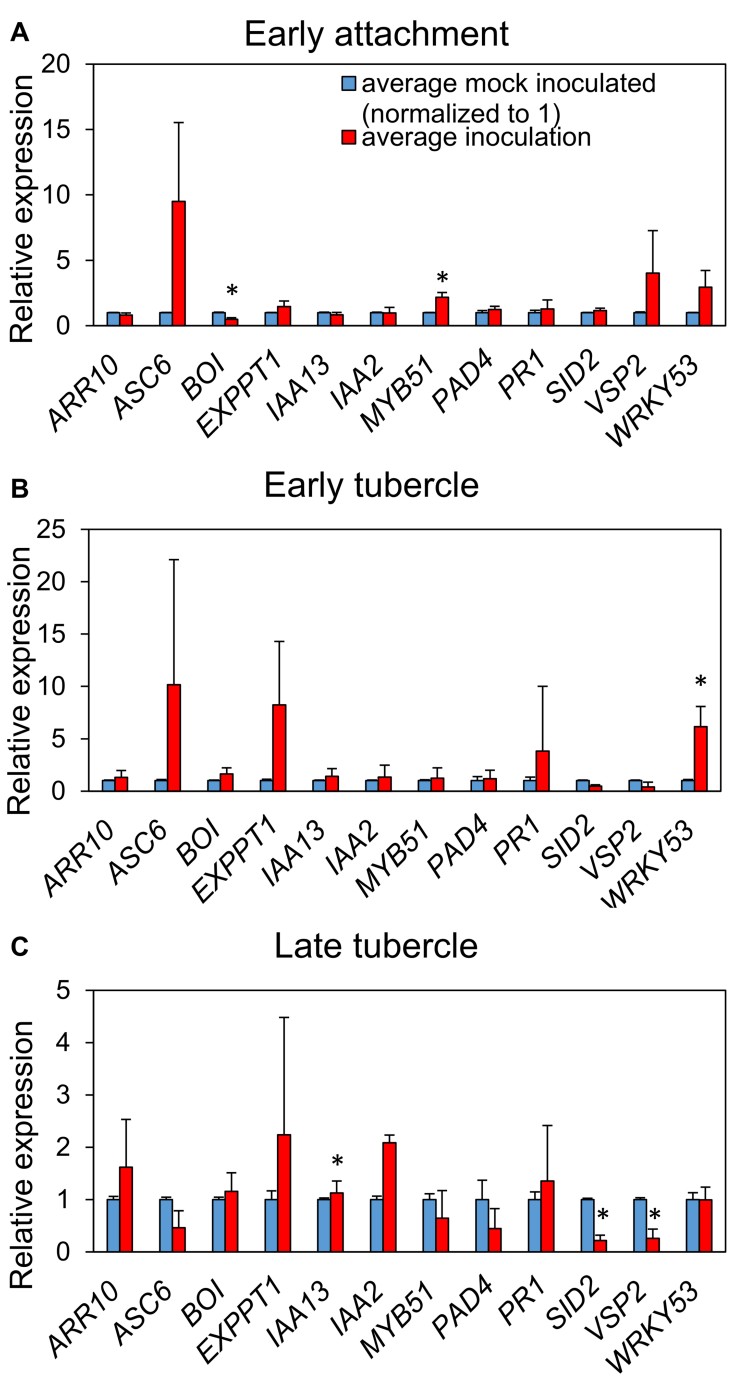

**Figure 5 *P. aegyptiaca* parasitization alters the transcription of marker genes for multiple hormone-signaling pathways.** Relative expression of twelve genes of interest were determined in the roots of both inoculated and mock-inoculated *A. thaliana* eco. Col-0 plants across the three studied stages: early attachment (A), early-stage tubercle development (B) and late-stage tubercle development (C). Data bars depict the expression of the 12 indicated genes relative to three stably-expressed house-keeping genes (*SAND, UBQ10* and *UFP*) from three biological and five technical replicates. Error bars represent the standard error. Data were normalized to set the relative expression of each gene in the mock-inoculated condition to one. Asterisks indicate statistically different means compared to mock-inoculated plants at the same growth stage based on paired *t*-test (*p* < 0.05).

we conclude that aberrations in host endogenous JA signaling can compromise compatibility to *P. aegyptiaca* parasitization. Further supporting this hypothesis, JA signaling genes were previously identified as upregulated in response to compatible parasitization of *Lotus japonicus* by *P. aegyptiaca* (*Hiraoka, Ueda & Sugimoto, 2008*), *A. thaliana* by *O. ramosa* (*Dos Santos et al., 2003*), sorghum by *Striga* (*Hiraoka & Sugimoto, 2008*) and *Medicago truncatula* by *Orobanche crenata* (*Die et al., 2007*). The common interpretation of these studies is that hosts induce JA-related defense responses to deter the parasite, but all of these hosts are susceptible to the respective parasites. Exogenous jasmonate treatment was previously shown to limit *P. aegyptiaca* parasitism (*Bar-Nun & Mayer, 2008*), suggesting that general perturbations in JA signaling may attenuate the compatibility of host plants to root parasitization.

Jasmonic acid signaling plays a critical role in numerous components of plant response to pathogen attack, including antagonism to SA signaling (*Gutjahr & Paszkowski, 2009*; *Kazan & Manners, 2008*). A reasonable hypothesis is that the JA mutants are less susceptible to parasitism because of elevated SA levels. High levels of SA are often associated with reduced virulence of biotrophic plant pathogens (*Thomma et al., 2001*). Application of exogenous SA was previously shown to reduce parasitism of clover by the closely related parasite *Orobanche minor* (*Kusumoto et al., 2007*). However, several SA-deficient mutants (e.g., *sid2* and *NahG*) were not more susceptible to parasite attachment and development. Additionally, a few mutants with known defects in SA accumulation and/or signaling (*ndr1* and *pad4*) were actually less susceptible to early-and late-stage tubercle development at the 95% confidence interval. Taken together, these results suggest that perturbations to functional crosstalk between SA and JA signaling limit the ability of *P. aegyptiaca* to parasitize *A. thaliana*. Therefore, we hypothesize that wildtype-functional SA and JA signaling is required by the parasite, potentially because these pathways are precisely targeted and manipulated during parasitism to support the attachment and development of the parasite. This hypothesis is further supported by the observation that the DELLA quad mutant plants are significantly less susceptible to late-stage tubercle development. The DELLA quad mutant was previously shown to be much more susceptible to biotrophs, likely due to aberrations in SA–JA crosstalk (*Navarro et al., 2008*). The DELLA genes are repressed by GA and parasitized tissue likely contains low levels of GA based on the expression of the marker gene *EXP-PT1* (Fig. 5).

## Putative hubs and regulators of plant immunity

*pfd6* plants were significantly more susceptible to *P. aegyptiaca* parasitization than wildtype host plants (Figs. 2–4). Even though our assay is biased toward identifying mutant host plants that are less susceptible to *P. aegyptiaca* parasitization, the increased susceptibility of *pfd6* was one of the strongest and most consistent phenotypes identified. PFD6 is a putative immunity hub protein (*Mukhtar et al., 2011*) involved in microtubule dynamics (*Gu et al., 2008*). It is possible that microtubules are being deployed to prevent parasitization, and disruption of microtubule formation is sufficient to support increased parasitization. Several plant pathogens target microtubules as part of

 

their attack (*Hardham, 2013*; *Lee et al., 2012*). Overexpression of *PFD6* or genes involved in similar functions may be sufficient to increase the resistance of host plants against parasitic plants.

Obligate holoparasites, such as *P. aegyptiaca*, are analogous to biotrophic microbial plant pathogens because they rely on living host tissue for the completion of their lifecycle. Multiple mutants that have hyper-activated immunity and are resistant to several biotrophic plant pathogens (*cpr5* and *edr1*) were substantially more resistant to *P. aegyptiaca* as well. This finding suggests that increased activation of known immune pathways that function in the control of biotrophic microbial plant pathogens is sufficient, in some instances, to also limit the severity of infestation by parasitic plants.

### Auxin/cytokinin

The *arr1/10/11* triple mutant supported more development of late-stage *P. aegyptiaca* tubercles than wildtype *A. thaliana* (Fig. 4) but only at the 95% confidence level. This finding suggests that host plant insensitivity to cytokinins leads to increased parasite attachment. The related parasite *Phtheirospermum japonicum* was recently shown to translocate cytokinins into host tissue to alter host development (*Spallek et al., 2017*). The reduced parasite attachment rates to *axr1* plants (Figs. 3 and 4) potentially contradict the hypothesis that altered cytokinin signaling enhances host susceptibility to *P. aegyptiaca*. Though initially identified as an essential component of *Arabidopsis* response to auxin treatment, AXR1 is also involved in sensitivity to ethylene (*Timpte et al., 1995*), cytokinin (*Li, Kurepa & Smalle, 2013*) and methyl-jasmonate (*Tiryaki & Staswick, 2002*). *axr1* mutant plants are expected to have lower sensitivity to cytokinins due to increased stability of the ARR5 protein, a negative regulator of cytokinin responses (*Li, Kurepa & Smalle, 2013*). We hypothesize that the reduced susceptibility to parasitization of the *axr1 A. thaliana* genotype is due to reduced sensitivity to ethylene (see below) as opposed to reduced sensitivity to cytokinins. Alternatively, reduced sensitivity to auxin signaling may explain the reduced susceptibility of *axr1* mutants to *P. aegyptiaca*. Local auxin biosynthesis and response gene activation were recently implicated as essential for haustorial formation by the *P. japonicum* (*Ishida et al., 2016*). However, the auxin signaling mutants *tir1* and *aux1* were not differentially susceptible to formation of *P. aegyptiaca* tubercles.

### Ethylene signaling

The ethylene signaling mutant lines, including *ein2*, did not support differential rates of *P. aegyptiaca* attachment and development compared to wildtype *A. thaliana*. Nevertheless, several observations suggest that ethylene signaling is an important part of host plant responses to *P. aegyptiaca* parasitism. First, host plants overexpressing *Erf2*, were substantially less susceptible to initial haustorial attachment by *P. aegyptiaca* (Fig. 2). Second, the *dde2* mutant line (JA biosynthesis) exhibited substantially reduced susceptibility to *P. aegyptiaca* tubercle development, but the quad signaling mutant line (*Tsuda et al., 2009*) (*dde2/pad4/sid2/ein2*) was not less susceptible to parasitization. This result suggests that loss of function in SA signaling or ethylene signaling is sufficient

to rescue the reduced susceptibility associated with defective host plant JA biosynthesis. Wildtype level *P. aegyptiaca* attachment rates were not rescued when infecting the double mutants *dde2/pad4* and *dde2/sid2* suggesting that the rescued attachment rate is due to the *ein2* mutation in the *dde2* background. We were not able to directly test this hypothesis because *dde2/ein2* double mutant plants do not grow in the PE bag system for unknown reasons. Third, the resistant-to-parasitism mutant *edr1* has perturbed ethylene signaling, specifically in the crosstalk of SA and ethylene signaling (*Tang, Christiansen & Innes, 2005*). However, the *edr1* phenotype may be more predominant due to EDR1's role as a negative regulator of plant immunity (see above). Given the high degree of crosstalk between JA and SA signaling with ethylene signaling (*Koornneef & Pieterse, 2008*), it will be challenging to determine the extent to which both of these host hormone-signaling pathways are independently required for successful *P. aegyptiaca* parasitization.

## RNAi

*RDR6* is involved in RNAi signaling and *rdr6* mutant plants are less susceptible to late-stage *P. aegyptiaca* tubercle formation, suggesting that *P. aegyptiaca* may exploit host-derived RNAi pathways during parasitization. The distantly related parasitic plant *Cuscuta campestris*, for example, secretes sRNA molecules into host plant tissue to silence plant-immunity and auxin-signaling gene pathways (*Shahid et al., 2018*). However, other mutants deficient in RNAi signaling (*dcl1* and *rdr1/2/6*) are not less susceptible to *P. aegyptiaca*. Indeed, both *dcl1* and *rdr1/2/6* are among the most susceptible mutants tested in terms of late-stage tubercle development (Fig. 4). DCL1 is essential for most miRNA biogenesis (*Bologna & Voinnet, 2014*) and plays a role in the negative regulation in an RDR6-involved RNA-silencing pathway (*Qu, Ye & Morris, 2008*). It is possible that *P. aegyptiaca* exploits this RNA-silencing pathway during parasitization, hence *dcl1* (negative regulator) mutants are highly susceptible and *rdr6* mutants are significantly less susceptible to tubercle development. However, the differential susceptibility of the *rdr6* and the *rdr1/2/6* lines cannot be explained at this time. A more thorough examination is necessary to determine the role of RNAi in *Arabidopsis*–*P. aegyptiaca* interactions.

## Pathways that appear not to be involved in susceptibility/resistance to *P. aegyptiaca* parasitization

It is important to note that the majority of tested mutant *A. thaliana* genotypes were not significantly different from wildtype in their susceptibility to *P. aegyptiaca* parasitization (Table S4). This result suggests that the ability of *P. aegyptiaca* to parasitize host plant tissue is robust, which is also supported by the relatively broad host range of *P. aegyptiaca* (*Parker, 2013*). Surprisingly, every tested *A. thaliana* mutant line was able to support some level of parasite attachment. In contrast, a recent similar screen of the obligate stem parasite *Cuscuta reflexa* on multiple tomato introgression lines revealed multiple genotypes where the host-parasite interaction was incompatible (*Krause et al., 2018*). We consider the lack of increased susceptibility of *bak1-4* plants to be the most surprising

result. We originally hypothesized that this mutant was the best candidate for a genotype more susceptible to parasitization because BAK1 is critical for many components of PTI (*Chinchilla et al., 2009*), which is a critical part of the plant immune response against many other classes of plant pests (*Jones & Dangl, 2006*) including the parasitic plant *C. reflexa* (*Hegenauer et al., 2016*). Future studies with the *bak1-5* allele that specifically blocks innate immunity without impairing brassinosteroid signaling (*Schwessinger et al., 2011*) may be informative. *mpk4* plants, which are deficient in map kinase signaling—another critical component of PTI (*Rasmussen et al., 2012*), were more susceptible to early attachment than wildtype (Fig. 2). But the increased susceptibility was not persistent through early and late stage *P. aegyptiaca* tubercle development. It is possible that *P. aegyptiaca* is already highly adept at subverting *A. thaliana* PTI. However, we show that the PTI marker gene *WRKY53* is significantly upregulated in response to tubercle development (Fig. 5). Several of the borderline increased susceptibility genotypes, such as *bak1-4*, should be tested in the PE bag system using a related parasite such as *O. minor* that is largely incompatible on *A. thaliana* (*Goldwasser, Plakhine & Yoder, 2000*). Starting with an incompatible interaction will more readily enable discovery of host genes that underpin resistance to parasitism than the assay described here performed with the compatible *P. aegyptiaca* parasite.

## CONCLUSION

This work demonstrates that the parasitization potential of *P. aegyptiaca* on *A. thaliana* is remarkably robust and not fully dependent on any of the genetic pathways tested here. However, *P. aegyptiaca* parasitization is at least partially limited in a small selection of *A. thaliana* mutant lines. The most robust phenotype revealed from this study is that multiple genes involved in JA signaling and biosynthesis are critical for full host plant susceptibility to *P. aegyptiaca*. Additionally, this work revealed that the putative immunity hub protein PFD6 is a critical component of the plant immune response that limits the severity of *P. aegyptiaca* parasitization. Further investigation into the identified genes that affect the degree of host plant susceptibility to parasitization will further elucidate the molecular mechanisms of plant parasitism. Alteration of these genetic pathways has the potential to help control parasitic weed infestation through either enhanced expression of resistance-associated genes or reduced expression of susceptibility-associated genes. Several of the susceptibility associated genes identified in this work cannot be knocked out without substantial negative impacts on the plants such as male sterility (e.g., *dde2*) or growth retardation (e.g., *cpr5*). This work serves as a starting point for understanding which genetic pathways are essential for plant susceptibility. Additionally, the viability of overexpressing *PFD6* or other identified resistance-associated genes to control parasitic plant infection needs to be determined in future work.

## ACKNOWLEDGEMENTS

We sincerely thank the Arabidopsis Biological Resource Center, John McDowell, Boris Vinatzer, Jeff Dangl, Ryan Anderson, Peter Morris, Nick Harberd, Georg Felix, Birgit Kemmerling, and Christian Voigt for providing seeds.

### Funding

This project was supported by the National Institute of Food and Agricultural through postdoctoral fellowship award 2015-67012-22821 to Christopher Clarke and award 135997 to James Westwood. Additional support was through the US National Science Foundation (IOS-1238057) to James Westwood. The funders had no role in study design, data collection and analysis, decision to publish, or preparation of the manuscript.

### Grant Disclosures

The following grant information was disclosed by the authors:
National Institute of Food and Agricultural: 2015-67012-22821 and 135997.
US National Science Foundation: IOS-1238057.

### Competing Interests

The authors declare that they have no competing interests.

### Author Contributions

- Christopher R. Clarke conceived and designed the experiments, performed the experiments, analyzed the data, prepared figures and/or tables, authored or reviewed drafts of the paper, and approved the final draft.
- So-Yon Park performed the experiments, prepared figures and/or tables, authored or reviewed drafts of the paper, and approved the final draft.
- Robert Tuosto performed the experiments, prepared figures and/or tables, and approved the final draft.
- Xiaoyan Jia performed the experiments, prepared figures and/or tables, and approved the final draft.
- Amanda Yoder analyzed the data, authored or reviewed drafts of the paper, and approved the final draft.
- Jennifer Van Mullekom analyzed the data, authored or reviewed drafts of the paper, and approved the final draft.
- James Westwood conceived and designed the experiments, authored or reviewed drafts of the paper, and approved the final draft.

### Data Availability

Data is available at the U.S. Department of Agriculture: Clarke, Christopher R.; Van Mullekom, Jennifer H.; Westwood, James H. (2019). Data from: Multiple immune pathways control susceptibility of *Arabidopsis thaliana* to the parasitic weed *Phelipanche aegyptiaca*. Ag Data Commons. DOI 10.15482/USDA.ADC/1503694. Accessed 13 May 2020.

### Supplemental Information

Supplemental information for this article can be found online at http://dx.doi.org/10.7717/peerj.9268#supplemental-information.

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
