# Peer review of "Multiple immunity-related genes control susceptibility of Arabidopsis thaliana to the parasitic weed Phelipanche aegyptiaca"

_PeerJ, doi:10.7717/peerj.9268_

## Round 0.1 · original submission · Major Revisions

The reviewers have identified a number of major and minor issues. It is important to think about the validity of your conclusions and try not to overstate them. It is important to describe the statistical analysis in detail., and provide a careful description of the entire process - starting from data collection, processing pipeline and results interpretation.

Reviewer 1 ·

Basic reporting

1. Goldwasser et al.2000 a and b is the same reference, similarly Tiryaki & Staswick 2002a and b;
2. I found it very difficult to read the graphs (Fig. 2- 4) and to draw a conclusion. It would easier if every pathway could be presented on a different graph or if not possible, maybe make it color coded to know in which pathways those mutants are involved; and make a line or cut off or mark somehow which mutants are less and which are more susceptible than WT.
3. Figure 5- there are some significance stars on the graphs but it is not mentioned in the legend or text how the statistic was done.
4. The time-lapse movie legend states that the experiment was done in Arabidopsis but for the above-ground growth tomato plants were used- this should be clarified in the movie description.

Experimental design

1. Some of the mutants are wrongly described, for instance jar1 is involved in JA biosynthesis and not signaling
2. the qPCR data is not convincing and not really discussed in the manuscript however a big conclusion is drown in the abstract. It would be better to use several marker genes for each pathway. Also the way the statistical analysis was done is not indicated.
Line 273-276: "The largest magnitude changes in measured gene expression were upregulation in ethylene (ACS6), JA (VSP2), and Pattern- triggered immunity (PTI, WRKY53)".All these genes were not significant while the significant ones BOI and MYB51 are not discussed.
3. A very high concentration of GR24 (6,7uM) was used and still low germination of P. aegyptiaca seeds was observed. I find it a bit odd as I thought that these seeds are very responsive to strigolactones or maybe not the most appropriate germination stimulant was used?

Validity of the findings

Although the novel quantitative assay was developed to assess growth of Phelipanche aegyptiaca on the host A.thaliana, for me, on this stage the manuscript did not provide additional knowledge on which pathways are important for the parasite-host interaction. According to me, the authors are not very critical about their data and often make too strong and far-reaching conclusions. For instance, they claim that functional JA biosynthesis and signaling pathways are critical for full susceptibility to P. aegyptiaca parasitization, while only dde2, JA biosynthesis mutant is less susceptible not jar1 and not signaling mutants jaz3/jaz4-1/jaz9-1 or OE Jaz3 (T3), jin1-1 at early tubercle and jaz3/jaz4-1/jaz9-1 mutant get less susceptible at late tubercle. I suggest to more carefully and criticaly reanalyze the data from mutants screen and to compare it with the qPCR results.
Also for pdf6 mutant, I agree that it is more susceptible than WT, but I would not make that strong conclusion that PFD6 is the immunity hub protein as its role in plant immunity was not studied in much detail.

Reviewer 2 ·

Basic reporting

The submitted manuscript by Clarke et al. with the title “Multiple immune pathways control susceptibility of Arabidopsis thaliana to the parasitic weed Phelipanche aegyptiaca” contributes new knowledge to the field of plant-parasite interaction. Phelipanche aegyptiaca, Egyptian broomrapes, is a parasitic plant that causes yield losses in tomatoes, carrots, and other crops (see Parker et al., 2012). Phelipanche infestations are notoriously difficult to manage, and genetic resistance is unknown.

Clarke et al. studied Arabidopsis thaliana as a host, rather than an economically important crop species. Working with the model species has the advantage of access to a large number of well-characterized mutants that can be shared between laboratories and tested with other parasites. Besides, working with Arabidopsis, a relatively small plant, is space-, time- and cost-effective. In line with previous studies, Clarke et al. show that at least early infection structures on Arabidopsis are comparable to other hosts (Westwood 2000, Bar-Nun et al.,2008).

The authors introduce in their manuscript a new quantitative assay for assessing the susceptibility of Arabidopsis towards P. aegyptiaca. The assay and the subsequent screening procedure of 46 Arabidopsis mutant lines are well-designed and well-documented. The authors provide sufficient background information, also, for readers of related research fields, e.g., plant immunity, plant hormone signaling. The manuscript is well structured, all figures, and movies, are relevant and of high quality. Raw data are accessible, as well as further in-depth documentation of the statistical analysis.

Experimental design

The authors discovered that Arabidopsis mutants with defects in the biosynthesis and the perception of the plant hormone jasmonate (JA) tend to support less P. aegyptiaca tubercle growth. In particular, the dde2-2 mutation negatively affected tubercle development in wild-type, sid2, and pad4 background, however not significantly in an ein3/sid2/pad4 genetic background. Similarly, the jar1 mutation that leads to reduced JA signaling in host plants negatively affect early and late tubercle development.

The other main discovery by Clarke et al. is that a mutation in the pfd6 Arabidopsis genes was linked to significantly more young tubercles, thus, indicating higher susceptibility. Pfd6 has not been studied previously in the context of parasitic plants. Pdf6 regulates immunity to bacterial pathogens through MTI, Microbe-associated molecular pattern-Triggered Immunity (Mukhtar et al., 2011). Clarke et al. study provide further evidence that parasitic plants trigger similar immune responses as microbial pathogens. An activation of MTI-like processes in Arabidopsis during Phelipanche infections was also seen in the up-regulation of the MTI-responsive gene WRKY53 in Figure 5.

Validity of the findings

In summary, the presented work by Clarke et al. is novel, exciting, and relevant, and I would like the authors only to consider the following minor points for revision:

Line 60: CuRe1 in tomato may be included as an example for a resistance gene detecting parasitic plants (Hegenauer et al., 2016) unless the authors refer to parasitic plants of the Orobanchaceae family only.

Line 109: Space missing in front of “mm”

Line 190: The authors wrote that their assay measures “attachment, vascularization, and tubercle growth.” The term “vascularization” is misleading here, as the assay did not assess the formation of vascular tissue, e.g., by stains or tracers per se, but uses tubercle growth as a proxy for successful vascular connections to host roots.

Figure 1B: “2 mg/ml” does not correspond to 6.7 µM GR24 (line 96).

Figure 1D: Please indicate on the x-axis or in figure legend if “% attachment” shows attachment relative to germinated seed or attachment relative to all seeds (germinated + un-germinated seeds).

Table S3: The aberrations “Quad sig” (likely) for dde2/pad4/sid2/ein2 is not explained in the text.

---

## Round 0.2 · Minor Revisions

There are several small items left before the paper is in acceptable shape. Both reviewers were concerned about the values of the concentration of GR24.

Reviewer 1 ·

Basic reporting

This manuscript is now improved in response to the comments from both reviewers. It is clearly written and overstated conclusions were removed.

Experimental design

1. I found one more mutant wrongly described in Table 1: sid2 is involved in SA biosynthesis not signaling.

2. I'm not sure whether paired t test can be used here for the qPCR data analysis as gene expression is compared between mock and treated plants and not twice for the same plant.

3. The concentration of GR24 was changed from 6,7uM to 2 mg/ml, which according to my calculation gives 6,7 mM so even higher than mentioned before. This should be clarified and maybe better to express it as molar concentration. Also the name "rac-GR24" instead of just "GR24" is now commonly used in the field.

Validity of the findings

All of my previous concerns were addressed.

Additional comments

The authors identified genes involved in resistance or susceptibility to P. aegyptiaca parasitization providing new knowledge to the field of plant-parasite interaction and giving a foundation for further studies.

Reviewer 2 ·

Basic reporting

no comment

Experimental design

The authors have satisfactorily addressed most of my questions. However, I am still confused about the actual concentration of GR24 that was used to pre-germinate P. aegyptiaca seeds. 2.2 mg/ml (~ 6.7 mM) would actually be three orders of magnitude higher than the previously stated 6.7 µM and also far above the range at which these germination inducers are applied in most studies. Maybe, there is only confusion about the final concentration in the Petri dishes, or perhaps this high concentration of GR24 was necessary to ensure homogenous germination. Either way, it is essential to clarify this point as it is a critical factor for any researcher who would like to conduct similar experiments.

Validity of the findings

no comment

---

## Round 0.3 · accepted · Accept

You have successfully addressed the comments, fixed all typos and mistakes. Therefore, I recommend acceptance